# Inhibition of Estrogenic Response of Yeast Screen Assay by Exposure to Non-Lethal Levels of Metallic Nanoparticles

**Byoung-cheun Lee, Cuong N. Duong, Jungkon Kim \*, Suejin Kim, Ig-chun Eom and Pilje Kim**

Risk Assessment Division, National Institute of Environmental Research, Incheon 22689, Korea;
skychen@korea.kr (B.-c.L.); ngoccuongsdg@sdglife.com (C.N.D.); suenier@korea.kr (S.K.);
iceom@korea.kr (I.-c.E.); newchem@korea.kr (P.K.)
**\*** Correspondence: jungkon@korea.kr; Tel.: +82-042-605-7052

**Abstract:** In order to investigate the effects of metallic nanoparticles (NPs) on the performance of in vitro bioassay, zinc oxide NP (ZnO NP), aluminum oxide NP ($Al_2O_3$ NP), bare silver NP (Ag NP), and Ag NP capped with citrate ($Ag^{cit}$ NP) were evaluated with yeast (*Saccharomyces cerevisiae* Y190) two-hybrid system (YES assay), carrying Japanese medaka estrogen receptors (mERs) in the presence of 17β-estradiol (E2, $10^{-6}$ M), a reference chemical for estrogenic activity. The distribution of NPs in the yeast was also examined by field-emission transmission electron microscopy (FE-TEM). The results show that TEM analysis revealed that NPs were present inside the yeast and accumulated deep inside the cell organelles, suggesting that cell death was caused by NPs. However, despite no significant change of mortality, the E2 estrogenic activities in yeast exposed to ZnO NP and $Al_2O_3$ NP were dose-dependently reduced. For Ag NP and $Ag^{cit}$ NP, such phenomenon observed in the exposure of ZnO NP and $Al_2O_3$ NP did not occur. From the observations, we found that ZnO NP and $Al_2O_3$ NP in the environmental media could result in underestimated estrogenicity of endocrine-disrupting compounds when evaluated by YES assay.

**Keywords:** nanoparticles; estrogen receptor; Japanese medaka; receptor binding assay; yeast

---

## 1. Introduction

In environmental monitoring, instrumental analyses using the chromatography technique is the best choice, but it is also a laborious, expensive, and time-consuming method. There is a growing need for rapid screening methods which can be applied at the previous state of the instrumental analysis. In vitro bioassay is currently considered as screening tool in a tiered approach, but there is still uncertainty about the reliability in environmental monitoring, because the results of in vitro bioassay may be affected by unidentified factors in the field samples [1].

The tiny size of nanoparticles (NPs), along with their substantially different characteristics, has led to their increased manufacture and application in commercial products. The increasing use of nanoproducts, such as in dental and skin care, self-cleaning textiles, drug development, information and communication technologies, and water treatment, makes organisms vulnerable to NP exposure [2–4]. The cellular toxicity of metal-based NPs has attracted the attention of many researchers around the world. Several studies have reported that metal-based NPs' treatment of non-lethal levels induce the adverse effects, such as mitochondrial dysfunction, membrane damage, intracellular oxidative stress, and abnormal cell morphologies [5–12], which alter specific functions of the cell.

Endocrine-disrupting compounds (EDCs) comprise a wide range of natural and synthetic compounds that can mimic or interfere with the functions of endocrine systems, regulating a variety

of biological functions in vertebrates, such as growth, metabolism, cell growth and proliferation, cell function and differentiations, sexual development and behavior, and development of the immune system in both sexes [1,13–15]. The yeast two-hybrid system (YES assay) has been wildly utilized to screen estrogenic compounds in environmental media. Considering NPs' unique physicochemical properties, including their high surface area relative to their volume, high interface energy, and high surface-to-charge ratio density [5], various interactions between NPs and EDCs in the water environment can be predictable. However, there has been no study about the effects of NPs in the environmental media on the results of YES assay. Additionally, little is known about the effects of NP exposure at the non-lethal level on the yeast, although the lethal toxicity of metallic NPs on the yeast has been well documented [16]. In the present study, we investigated the effects of metallic NPs on the performance of YES assay with a Japanese medaka estrogen receptor (mER).

## 2. Materials and Methods

### 2.1. Chemicals and Test Preparation of NP Solutions

Ag NP (<150 nm, Lot # 06412DE), ZnO NP (<100 nm, Lot # MKBJ7906V), $Al_2O_3$ NP (<50 nm, Lot # BCBG4586V), dimethyl sulfoxide (DMSO), 17β-estradiol (E2), and amino acids needed for cell culture were purchased from Sigma Aldrich (Steinheim, Germany). Citrate-capped Ag NP ($Ag^{cit}$ NP) was obtained from ABC Nanotech (Daejon, Korea). Yeast extract, agar, dextrose, and peptone were purchased from Bacto Laboratories Pty Ltd. (Mt Pritchard, NSW, Australia). Metallic NP stock solutions were prepared in deionized (DI) water by vigorously stirring for 24 h with a shaking machine and were stored in the dark until the beginning of experiments. Working solutions were obtained by serial dilution of stock solutions in DI water prior to the tests. To prevent aggregation of NPs, stock solutions were sonicated for at least 10 min before dilution. An aliquot of the prepared working solution was immediately transferred to quartz cuvettes, and the hydrodynamic sizes of NP dispersions were measured using a dynamic light scattering (DLS) method with an ELS-Z-2 analyzer (OTSUKA, Japan). Solution pH was unadjusted, all measurements were carried out at least in triplicate, and average values were reported.

### 2.2. Estrogenic Activity Assay

A yeast stock, obtained from Dr. Fujio Shiraishi at the National Institute for Environmental Studies, Japan [17] and stored at −80 °C, was thawed to room temperature before mixing with modified synthetic defined (MSD) medium (Table S1). A YES assay, using the yeast (*Saccharomyces cerevisiae* Y190) transformed with an mER, was used to evaluate estrogenic activity [18–20]. The assay was adapted to a chemiluminescent reporter gene method employing a 96-well culture plate. Yeast cells were preincubated for 24 h at 30 °C with shaking in MSD medium, without tryptophan and leucine, and the cell density adjusted to an absorbance of 1.75 to 1.85 at 595 nm. E2 and/or NPs for test were serially diluted and placed in the wells of a black 96-well culture plate. Then, the yeast cell suspension was also added to each well. After the addition of the yeast suspension and vortex mixing, the plates were incubated at 30 °C for 4 h. Yeast cells were lysed using a lysis solution containing 70% zymolyase-100T in Z-buffer and 30% reaction buffer (AURORA GAL-XE; MP Biochemical, Solon, OH, USA). The plate was incubated at 37 °C for 1 h and then placed in a 96-well plate luminometer. The chemiluminescence produced by β-galactosidase in each well was measured. Detailed compositions of the media, agar, and other buffers are listed in Table S1.

Dose–response relationship of E2 was established by exposing yeast to a series of E2 concentrations from $10^{-11}$ to $10^{-4}$ M. We found $10^{-6}$ M of E2 was close to the concentration of median estrogenic activity (EC50) (Figure 1). In order to measure the change of estrogenic activity by NPs, 60 μL of yeast solution and 60 μL of test solutions containing NPs and the EC50 level of E2 ($10^{-6}$ M) were added to each well. The changes in estrogenicity by NPs were expressed in the ratio of the activity level of only $10^{-6}$ M E2 exposure. The concentration range of ZnO NP, $Al_2O_3$ NP, and $Ag^{cit}$ NP was 0.2 mg/L,

1 mg/L, 5 mg/L, 10 mg/L, 50 mg/L, and 100 mg/L, whereas that of Ag NP was 0.05 mg/L, 0.1 mg/L, 0.5 mg/L, 1 mg/L, 5 mg/L, and 10 mg/L, respectively. Two experiments were independently carried out, and for each test concentration, three replicates were used.

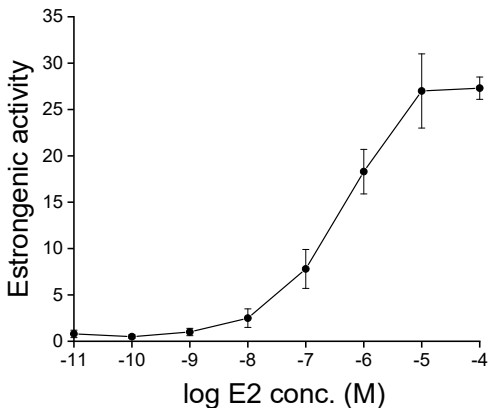

**Figure 1.** Dose–response relationship for estrogenic activity in yeast exposed to E2.

## 2.3. Viability Test

Viability tests were conducted in parallel with the estrogenic activity test. After each test, 10 μL of test solution was removed for a viability test using a staining method with 0.4% Trypan blue (Sigma-Aldrich, St. Louis, MO, USA). Diluted yeast solutions were mixed with Trypan blue at a ratio of 1:5 and incubated at room temperature for 10 min. The cells were then observed under a light (Leica DMI 3000B, Leica Microsystems GmbH, Wetzlar, Germany). Dead yeast cells were recognized by their blue color, whereas living cells appeared bright and colorless with Trypan blue staining. The counting was conducted in triplicate, and the results were averaged. Two experiments were independently carried out, and for each test concentration, three replicates were used.

## 2.4. Transmission Electron Microscopy (TEM)

For TEM imaging, yeast cells exposed to E2 with NPs (50 mg/L ZnO NP, 50 mg/L $Al_2O_3$ NP, 50 mg/L Ag NP, 100 mg/L $Ag^{cit}$ NP) were primarily fixed with 2% paraformaldehyde and 2% glutaraldehyde in 0.05 M sodium cacodylate buffer (pH 7.2) at 4 °C for 4 h. After washing 3 times in 0.05 M sodium cacodylate buffer (pH 7.2) at 4 °C, cells were post-fixed with 1% osmium tetroxide in 0.05 M sodium cacodylate buffer (pH 7.2) at 4 °C for 2 h. The fixed cells, after a quick rinsing with DI water, were en bloc stained with 0.5% uranyl acetate at 4 °C for 12 h. The stained cells were dehydrated in a series of ethanol concentrations (i.e., 30%, 50%, 70%, 80%, 90%, and 100%). The processing steps were followed by infiltration using propylene and Spurr's resin for 16 h, followed by polymerization at 70 °C for 24 h. Finally, the pellets were sectioned using an ultramicrotome (MT-X; RMC, Tucson, AZ, USA) and observed under field-emission transmission electron microscopy (FE-TEM) (LIBRA 120; Carl Zeiss, Germany).

## 2.5. Statistical Analysis

Datasets were obtained in triplicate, calculated, and analyzed using the Excel® software (Microsoft Corporation, WA, USA) and plotted using Sigma Plot (SPSS Inc., San Jose, CA, USA). The results were reported as mean ± standard deviation. Analyses of variance (one-way ANOVA), followed by Dunnett's test as a post hoc test, were performed for the estrogenic activity test and the viability test. The differences were considered significant at $p$-values < 0.05.

## 3. Results

### 3.1. Particle Characterization

The sizes of the particles using the DLS method are shown in Table 1. The average sizes of the NPs suspended in DI water were (201.4 ± 30.7) nm, (147.9 ± 54.0) nm, (38.5 ± 10.1) nm, and (20.0 ± 1.2) nm, for ZnO NP, $Al_2O_3$ NP, Ag NP, and $Ag^{cit}$ NP, respectively, with monodispersed phase.

**Table 1.** Hydrodynamic sizes of nanoparticles (NPs) tested.

| Nanoparticles | Hydrodynamic Size (nm) |
|---|---|
| ZnO NP | 201.4 ± 30.7 |
| $Al_2O_3$ NP | 147.9 ± 54.0 |
| Ag NP | 38.5 ± 10.1 |
| $Ag^{cit}$ NP | 20.0 ± 1.2 |

Values in table represent mean ± standard deviation.

### 3.2. Dose–Response Relationship of Estrogenic Activity in Yeast Exposed to E2

Figure 1 shows the dose–response relationship of the estrogenic activity in yeast exposed to a series of E2 concentrations from $10^{-11}$ M to $10^{-4}$ M. The estrogenic activity was dose-dependently elevated upon exposure to $10^{-5}$ M E2, and reached a plateau. We found that $10^{-6}$ M of E2 was close to the concentration of EC50. Based on the dose–response relationship, $10^{-6}$ M of E2 was chosen for subsequent tests investigating the effects of NPs on the estrogenic activity of E2.

### 3.3. Effects of NPs on Viability of Yeast and Estrogenic Activity of E2

Figure 2 shows the effects of NPs on the viability and the activity of E2 in the yeast two-hybrid system. When compared to control, no significant difference was observed in the viabilities of yeast cells exposed to ZnO NP and $Al_2O_3$ NP, except for 100 mg/L (Figure 2a,b). However, E2 estrogenic activities began to decline at 5 mg/L of ZnO NP and 10 mg/L of $Al_2O_3$, respectively. With the 50 mg/L NPs, the estrogenic activities were about 40%, compared to the control. Ag NP did not induce any significant difference in the viability and the estrogenic activity of E2 in yeast cells up to a concentration of 0.1 mg/L (Figure 2c). However, with 0.5 mg/L Ag NP, the viability and estrogenic activity significantly decreased ($p < 0.05$) in comparison with control. For $Ag^{cit}$ NP, the mortality increase was slower than that with Ag NPs, and estrogenic activity was inhibited at a concentration of 10 mg/L and above (Figure 2d).

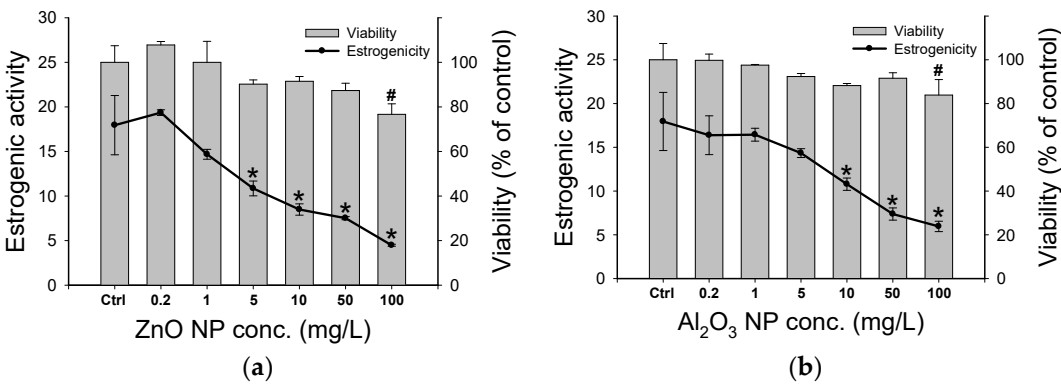

**Figure 2.** *Cont.*

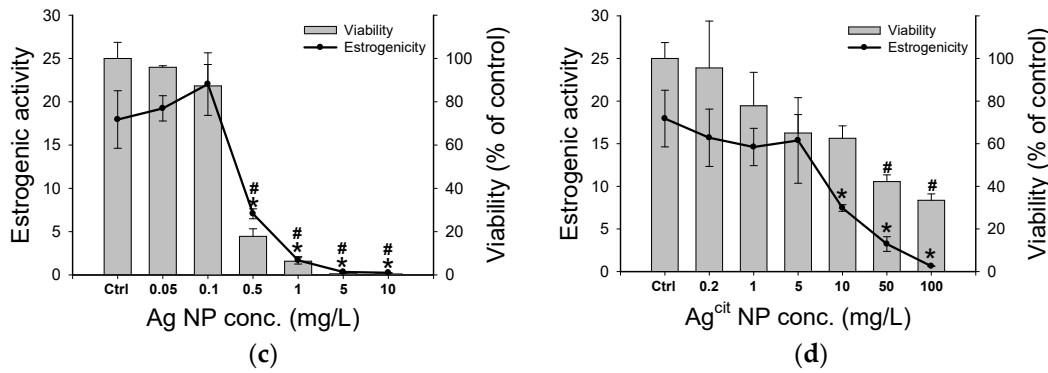

**Figure 2.** Viability and estrogenic activity of yeast exposed to (**a**) ZnO NP, (**b**) Al$_2$O$_3$ NP, (**c**) Ag NP, and (**d**) Ag$^{cit}$ NP, in combination with 10$^{-6}$ M E2. Asterisks for estrogenic activity test and pounds for viability test indicate a significant difference relative to the control ($p < 0.05$), using Dunnett's post hoc test.

### 3.4. Particle Distribution in Yeast and Cell Death

Figure 3 shows cellular morphologies of intact cell and damaged cell by exposure of E2 with NPs (50 mg/L ZnO NP, 50 mg/L Al$_2$O$_3$ NP 50 mg/L Ag NP, 100 mg/L Ag$^{cit}$ NP). Control cell was oval, with a clear and continuous membrane, whereas cell exposed to NPs exhibited the non-oval shape with the destroyed membrane. No particles were observed in the control cells. However, NPs penetrated and distributed evenly inside the cells, and also attached thickly on the cell membrane of yeast (Figure 3b).

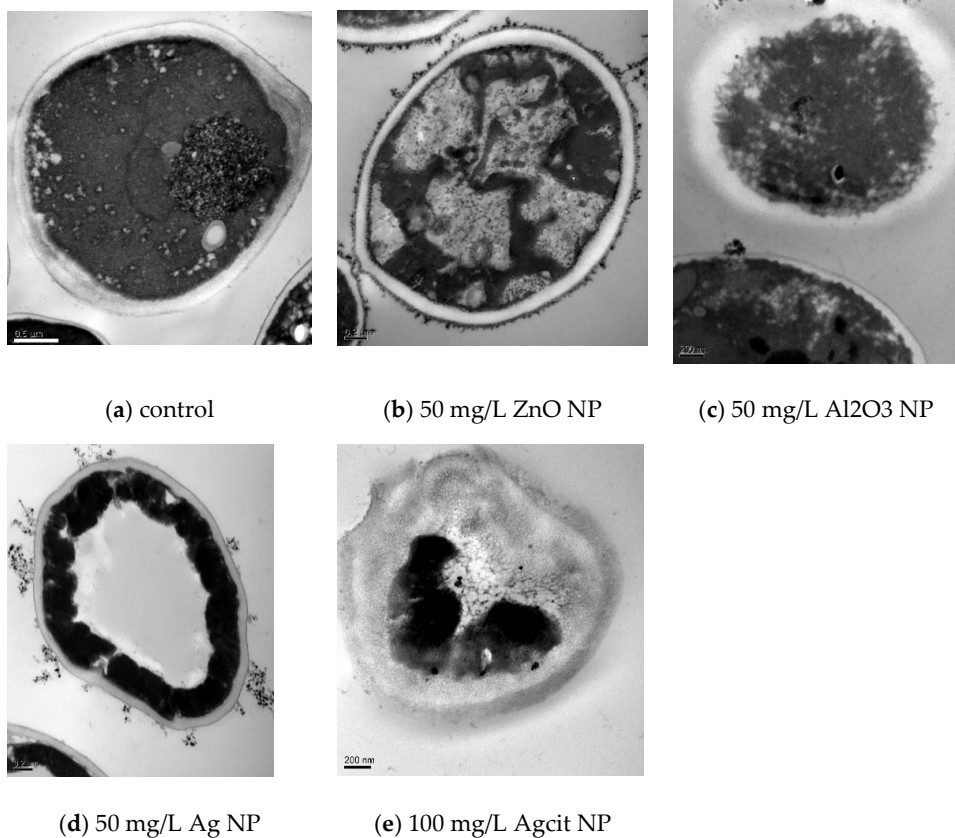

**Figure 3.** Representative transmission electron microscopic images showing distribution of nanoparticles in yeast.

## 4. Discussion

Although the NPs formed agglomerates and aggregates, single particles still existed and acted as a major source of particles inside the cells. TEM analysis revealed that ZnO NPs were present inside the yeast cells as single or small aggregates of particles (Figure 3b) and accumulated deep inside the nucleus or protoplasm. The penetration of NPs into the cells or their adhesion to the cell membrane results in membrane destruction, and the cells eventually die.

Despite no significant change in mortality, the E2 estrogenic activity in yeast exposed to ZnO NP was reduced as a function of the particle concentration (Figure 2a). Similar response patterns in mortalities and estrogenic activities were observed in the experiment using $Al_2O_3$ NP, suggesting a toxicity profile similar to that of ZnO NP (Figure 2b). Kasemets et al. [16] reported that ZnO NP and bulk ZnO both dose-dependently inhibited the growth of yeast. However, the effects of non-cytotoxic concentrations of ZnO NP still remain unknown. The toxicity of metallic NPs in mammalian cells, as well as the adverse effects, such as mitochondrial dysfunction, membrane damage, intracellular oxidative stress, and abnormal cell morphologies, has been previously reported [20–24]. From the results obtained in this study, it can be inferred that exposure to non-cytotoxic concentrations of NPs (5–50 mg/L for ZnO NP and 10–50 mg/L for $Al_2O_3$ NP) may lead to cell membrane damage. The damaged membranes, in turn, may incidentally accelerate the cellular uptake of NPs. The toxicity of given NPs may lead to cell death, or merely alter a specific function of the cell. Yeast two-hybrid system applied in this study is worked by biological multi-scale processes, such as ligand–receptor binding, protein–protein and DNA–protein interactions, gene expression, and enzyme reaction. If not cell death, the signal reduction of the YES assay indicates the hindrance of binding between E2 and receptor, or the malfunction of other biological processes mentioned above. It can be hypothesized that NPs' inner or outer cells reduce E2 levels, by means of the sorption between NPs and E2. A previous study showed that the E2-sorption capacity of particles was negligible [25], even when the particles were coated with humic acids. Therefore, reduction of available E2 levels by the sorption between NPs and E2, in this case, is unlikely.

It is worth noting that at lower concentrations (i.e., <0.5 mg/L), Ag NP stimulated the E2 estrogenic activities (Figure 2c). Although the E2 estrogenic activity was not significantly increased, a similar pattern was repeatedly observed in the tests using other metallic NPs. This phenomenon may be due to the inadequate destructive strength of Ag NPs to kill the yeast cells in a short time. This insufficient destructive strength may affect the cell membrane, resulting in the easier transport of exogenous chemicals, i.e., E2, into the cell. Finally, the activity of the indicator (i.e., β-galactosidase) increased within a short period. With increasing particle concentrations, more yeast cells were killed, and the estrogenic activity was consequently reduced. The high correlation coefficients between yeast viability and estrogenic activity confirm this proportional relationship, which also supports the above theory.

Many studies about NPs have, up to now, been done based on a precautionary approach, because of the lack of tools to detect and quantify NPs in the environment [26]. Kaegi et al. [27] have, however, succeeded in tracing the $TiO_2$ NP emitted from exterior paints in a small stream. This means that the NPs in the environment become real concerns. Many in vitro assays, including YES assay, are being used to screen the adverse effects of environmental contaminants. According to the present study, metallic NPs could disturb the processes of in vitro assays, resulting in underestimating the adverse effects of contaminants. Therefore, the effects of NPs should be taken into consideration when evaluating environmental samples by in vitro assay.

**Supplementary Materials:** The following are available online at http://www.mdpi.com/2076-3417/10/11/3796/s1, Table S1: Compositions of the media, buffers, and agar used in the experiments.

**Author Contributions:** Conceptualization: B.-c.L. and C.N.D.; methodology: B.-c.L. and C.N.D.; formal analysis: C.N.D. and J.K.; writing—original draft preparation: B.-c.L. and C.N.D.; writing—review and editing: J.K., I.-c.E., and P.K.; project administration: S.K. All authors have read and agreed to the published version of the manuscript.

**Funding:** This research was supported by grants from the National Institute of Environmental Research, funded by the Ministry of Education of Korea (11-1480523-000343-01).

**Conflicts of Interest:** The authors declare that here are no potential conflicts of interest with respect to the research, authorship, and/or publications of this article.

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
