# Peer review of "Inhibition of Estrogenic Response of Yeast Screen Assay by Exposure to Non-Lethal Levels of Metallic Nanoparticles"

_applsci, doi:10.3390/app10113796_

Round 1

Reviewer 1 Report

The manuscript entitled "Inhibition of estrogenic response of yeast screen assay by exposure to non-lethal levels of metallic nanoparticles" gives some interesting data concerning the biological response of metallic nanoparticles in yeasts. However, in my opinion, the manuscript needs minor improvements before it can be published in  Applied Sciences

1) Preparation of nanoparticles solution for characterisation needs more precise description.  If the stock solution of nanoparticles was prepared immediately before experiments or earlier?

2) DLS measurements need a more precise description. It is not clear how did the authors measure the size of nanoparticles (the type of cuvettes, pH of tested solutions, etc.). How long from the sonication, the size of NPs was measured? Why wasn't the zeta potential measured or the polydispersity index given? Both values are essential because they show the stability of the nanoparticles in solution. Whether the authors have studied the aggregation of nanoparticles in time? Metallic nanoparticles aggregate in solution, and it can affect the toxicity results.

3) What is the difference between Figure 3b and 3c? Both Figures are signed 50 mg/L ZnONPs, but t can be seen that they differ.

Author Response

On behalf of the all the coauthors, I appreciate your constructive comments that significantly improved the quality of this manuscript. I tried to answer to your questions and comments item by item. "Please see the attachment." 

Reviewer 2 Report

The authors reported a study investigating the effects of several metal nanoparticles on the performance in environmental monitoring of an in vitro bioassay, namely YES assay. They  showed that  metallic NPs could modify  the  processes of in vitro assays and concluded that the effects of NPs should be taken into consideration when evaluating environmental samples by in vitro assay. The paper could be considered for publication if taking into consideration the followings:

  1. Please, explain the abbreviation ‘NPs’ in line 11 not in lines 32-33 from introduction.
  2. Please, replace “physic-chemical” with “physico-chemical” (line 49).
  3. I suggest you to use the same abbreviation (NP) for all the compounds (line 58).
  4. In the section 2.4 you mentioned that you used TEM for ZnONP and AgcitNP but in the results section you presented the results for three types of nanoparticles. Please clarify and give explanation for Figure 3 c,d,e.
  5. In Figure 1 please use logM for E2 concentration.
  6. Figure 2 describe estrogenic activity not Figure 3 (lines 171-172). Please do the necessary corrections.

Author Response

(The authors gave the same response as above.)

Reviewer 3 Report

In this article, the authors used the yeast two hybridization assay to investigate the effects of non-lethal doses of metallic nanoparticles. The experimental design is appropriate and adequately tests the research question. Below are a few minor comments:

Comment 1: Line 11 “In order to investigate…. metallic NP……”  Use nanoparticle (NP) in the first sentence and then NP throughout.

Comment 2: Line 46 “To data, however….” Please reconstruct the sentence for clarification.

Line 49: “Physic-chemical properties” – please correct

In general, the introduction section does not flow well and hard to comprehend in some areas to tie everything to the question of interest.

Author Response

(The authors gave the same response as above.)
